# *Limosilactobacillus reuteri* M4-100 Mitigates the Pathogenicity of *Escherichia coli* Strain HMLN-1 in an Intestinal Epithelial Model and Modulates Host Cell Gene Expression

**DOI:** 10.3390/microorganisms13061428

**Published:** 2025-06-19

**Authors:** Behnoush Asgari, Georgia Bradford, Eva Hatje, Anna Kuballa, Mohammad Katouli

**Affiliations:** 1Centre for Bioinnovation, and School of Science, Technology and Engineering, University of the Sunshine Coast, Maroochydore, QLD 4558, Australia; basgari@usc.edu.au (B.A.); georgia.bradford@research.usc.edu.au (G.B.); 2Centre for Immunology and Infection Control, Faculty of Health, School of Biomedical Sciences, Queensland University of Technology, Brisbane, QLD 4000, Australia; e.hatje@qut.edu.au; 3Centre for Bioinnovation, and School of Health, University of the Sunshine Coast, Maroochydore, QLD 4558, Australia; akuballa@usc.edu.au

**Keywords:** translocation, *Escherichia coli*, intestinal epithelium, *Limosilactobacillus reuteri*, differential gene expression

## Abstract

Probiotics have been widely adopted due to their beneficial health properties. Here, we investigated the interactions of a probiotic *Limosilactobacillus (Lactobacillus) reuteri* M4-100, with a translocating *Escherichia coli* strain HMLN-1, in a co-culture of cells, representing the intestinal epithelium, and identified molecular mechanisms associated with the host response. A co-culture of Caco-2:HT29-MTX cells was exposed to the HMLN-1 strain and the route of translocation was studied. Scanning and transmission electron microscopy revealed the adhesion of the strain to the microvilli, the establishment of close contact with the co-culture prior to being taken up by membrane-bound vesicles, and translocation via the intracellular pathway. When the HMLN-1 strain was challenged with *L. reuteri* M4-100 in co- and pre-inoculation experiments, its adhesion to the co-culture of cells was significantly reduced (*p* < 0.0001). A significant reduction in the invasion of the HMLN-1 strain was also observed upon the inoculation of *L. reuteri* M4-100 with the co-culture 60 min prior to HMLN-1 exposure (*p* < 0.0001). The *L. reuteri* M4-100 strain also significantly (*p* < 0.0001) reduced the translocation of the HMLN-1 strain in both co- and pre-inoculation experiments. Differential gene expression studies identified key cellular responses to the interaction with these bacteria, both alone. These data demonstrate the efficacy of *L. reuteri* M4-100 to reduce or inhibit the interaction of *E. coli* HMLN-1 with the intestinal epithelium. A prophylactic role of this probiotic strain is postulated as these effects were more pronounced in pre-inoculation experiments.

## 1. Introduction

Bacterial translocation is the movement of live bacteria through the gut epithelium to the mesenteric lymph nodes (MLNs) and blood circulation to reach the otherwise sterile organs [1]. An increased number of post-operative septicemia cases has been reported to be positively associated with this process [2]. *Escherichia coli* is one of the leading causes of gut-associated septicemia despite having a low population size in the gut microbiota. These strains are commonly isolated from septicemic patients in hospitals [3,4,5] and are thought to be part of the commensal *E. coli* of the gut [6]. However, due to their specific ability to translocate, they are referred to as translocating *E. coli* (TEC) as their translocation ability has been shown to be host species-specific [7]. Among TEC strains, *E. coli* HMLN-1 was originally isolated from the blood and mesenteric lymph nodes of an individual with a fatal case of pancreatitis [8], and our previous studies have demonstrated its ability to efficiently translocate in animals [7,9,10] and in mono-culture cell lines representing intestinal epithelium [11]. Most studies investigating the translocation of gut bacteria have used monoculture cell lines, which do not fully represent intestinal epithelial properties due to a lack of mucin, one of the important barriers of the intestinal epithelium against pathogens [12,13,14]. In this study, we employed a co-culture of two cell lines, Caco-2 and HT-29 mutated with methotrexate (MTX), (HT29-MTX) capable of producing mucin [12,15] to investigate the effect of the HMLN-1 strain on eukaryotic cells.

The use of some lactobacilli strains as probiotics to support beneficial health outcomes has gained interest, particularly in combating microbial infections of the intestinal epithelium [16]. These strains offer a promising avenue for treatment by restoring balance to the gut microbiota and providing protective effects against harmful pathogens, without contributing to the problem of antimicrobial resistance [17,18,19,20]. We previously isolated four lactobacilli strains from well-defined healthy individuals and characterized them to the species level [21]. We further showed that these strains could inhibit the interaction of adherent–invasive *E. coli* (AIEC) with a co-culture of Caco-2:HT29-MTX cells with the *L. reuteri* M4-100 strain having a better ability to reduce adhesion and the invasion of the AIEC strain *E. coli* [22].

The aim of this study was to establish the ability of *L. reuteri* M4-100 to inhibit the adhesion, invasion, and translocation of a TEC strain (HMLN-1). We also aimed to investigate the route of translocation by which the HMLN-1 strain crosses the intestinal epithelium in our co-culture model. Additionally, we aimed to investigate the host cell response to HMLN-1 and *L. reuteri* M4-100 exposure by evaluating differential gene expression within the intestinal cell model.

## 2. Materials and Methods

### 2.1. Bacterial Strains

The *E. coli* HMLN-1 strain was originally isolated from the blood and MLNs of an individual with a fatal case of hemorrhagic pancreatitis [8] and has been tested for its ability to translocate in mono-culture cell lines representing the intestinal epithelium [11]. The strain was stored in tryptone soy broth (TSB) with 20% glycerol at −80 °C, and upon revival, the purity of the strain was tested through streaking on MacConkey agar no. 3 (Oxoid, Thebarton, Australia) and incubation at 37 °C for 24 h. Isolated colonies were then selected and grown on nutrient agar (Oxoid, Thebarton, Australia) as working cultures for cell culture assays and for nucleic acid extraction.

*L. reuteri* M4-100 was also preserved in de Man, Rogosa, Sharpe (MRS) broth (Oxoid, Thebarton, Australia) and 20% glycerol at −80 °C. The working culture of the strain was stored at 4 °C and grown anaerobically on MRS broth for 24 h at 37 °C for each assay. The strain suspension was then centrifuged at 3500 rpm for 5 min and resuspended in phosphate-buffered saline (PBS, pH = 7.4). The concentration of the probiotic strain was then adjusted to an optical density of 1.0 at OD_600nm_ (approx. 10^9^ CFU/mL) and adjusted to 10^7^ CFU/mL before inoculation of the wells.

### 2.2. Cell Culture Experiments

A co-culture of Caco-2 cells (ATCC HTB 37) and HT29-MTX (ATCC HTB 38) (Manassas, VA, USA) was used as a model of the intestinal epithelium to test the interaction of HMLN-1 with the intestinal epithelium as described previously [22,23]. Caco-2 cells were used due to their ability to differentiate into a polarized monolayer with tight junctions and a microvilli brush border [24,25]. However, they do not have the ability to produce a substantial level of mucin [25]. To overcome this, HT-29 cells were treated with methotrexate allowing them to differentiate into HT29-MTX goblet cells that secrete mucin [26]. HT29-MTX cells do not exhibit effective tight junction formation leading to a compromised monolayer; therefore, a co-culture of Caco-2 and HT29-MTX cell lines was used successfully [27]. By seeding Caco-2 and HT29-MTX cells, a monolayer develops that secretes sufficient amount of mucin whilst expressing tight junctions and microvilli brush borders, similar to that of the GI epithelium [28].

Cell lines were initially grown in Eagles Minimum Essential Media (EMEM) (Sigma Aldrich, Melbourne, Australia). The Caco-2 cells were supplemented with 20% fatal bovine serum (FBS) (Lonza, Sydney, Australia) whilst HT29-MTX cells were supplemented with 15% FBS for growth. The culture medium was supplied with 600 µL of penicillin–streptomycin (ThermoFisher, Brisbane, Australia). Cells were grown in 25 cm^2^ cell culture flasks and incubated at 37 °C in a 5% CO_2_ environment, and media were replaced every 48 h as described before [23]. Both Caco-2 and HT29-MTX cells were grown separately and seeded at a 9:1 ratio (Caco-2: HT29-MTX) in an 8-well chamber slide (Nunc Lab-Tek II, Thermofisher, Woolloongabba, Qld, Australia).

### 2.3. Adhesion Assay

For the adhesion assay, the co-culture was grown to >80% confluence and chamber slides were washed three times with PBS and the media replaced with antibiotic-free EMEM. The *L. reuteri* M4-100 strain was grown in MRS broth (Oxoid) overnight while the HMLN-1 strain was grown in LB broth (Merck) in a reciprocal shaker (138 strokes.min^−1^) at 37 °C for 24 h. After growth, the bacterial suspension was centrifuged (4000 rpm, 12 min) and the supernatant discarded. The bacterial pellet was resuspended in PBS (pH 7.4) to an OD_600nm_ of 1 (approximately 10^9^ CFU/mL). Inhibition of adhesion was performed in a co-inoculation assay by adding 100 µL of both *E. coli* HMLN-1 and *L. reuteri* M4-100 suspensions simultaneously into each chamber (co-inoculation) and then incubated at 37 °C under a 5% CO_2_ environment for 90 min. Pre-inoculation assays were conducted by adding 100 µL of *L. reuteri* M4-100 strain first for 90 min followed by inoculating the HMLN-1 strain and further incubation for 90 min. Slides were then washed three times with PBS to remove non-adhering bacteria. Chambers were then fixed with 95% ethanol for 5 min and Gram-stained for light microscopy observation. All adhesion assays were completed in duplicate. The ability of the HMLN-1 and *L. reuteri* M4-100 strains to colonize the gut epithelium was assessed as the percentage of cells showing adhesion after randomly counting 100 cells. The number of adhering bacteria per cell (CFU/cell) was calculated after counting the number of bacteria on 25 randomly selected cells showing adhesion. The results are expressed as the mean ± standard error of mean (SEM) of bacteria adhering per cell. The *E. coli* strain JM109 was used as the negative control.

### 2.4. Invasion Assay

Co-culture cells were grown onto a flat-bottom 96-well plate until full confluence. The medium was then removed, and the wells were washed three times with PBS and replaced with antibiotic-free EMEM. A similar procedure to the adhesion assay was used for the inoculation of bacteria. Both HMLN-1 and *L. reuteri* M4-100 were grown as described above. The same concentration of bacteria (1.0 × 10^7^ CFU/mL) was used to inoculate each well.

Inhibition of invasion in a co-inoculation assay was tested by inoculating the cells simultaneously with both *E. coli* HMLN-1 and *L. reuteri* M4-100 and incubated for 90 min at 37 °C in a 5% CO_2_ environment. The pre-inoculation assay was conducted by allowing *L. reuteri* to interact with the cells for 90 min before inoculating wells with the HMLN-1 strain and further incubating for 90 min. Wells were then inoculated with gentamicin (150 µg/mL) for 60 min to kill any extracellular bacteria, and the contents were removed and washed three times with PBS. Wells were then incubated with 0.1% Triton-X-100 (Sigma-Aldrich) for 15 min to lyse the monolayer releasing invading bacteria. The lysate was serially diluted and 100 µL aliquots were plated onto MacConkey agar no. 3 plates (Oxoid) and incubated overnight at 37 °C. Invasion assays were performed in duplicate. Colonies were then counted, and the CFU was expressed as mean ± SEM. *E. coli* strain JM109 was used as a negative control.

### 2.5. Translocation Assay

A co-culture of cells was grown in EMEM onto cell Millipore inserts (Millipore, Melbourne, Australia) with a permeable base containing 8 µm diameter pores and placed onto 24 well plates with EMEM with 1% penicillin–streptomycin. The co-culture cells were grown until a fully confluent monolayer was observed, which was indicated by a plateauing trans-epithelial electrical resistance (TEER) value (ohms = Ω) or in the range of 14–21 days of growth [29]. Cell inserts and wells were washed three times with PBS and replaced with antibiotic-free EMEM. The same concentrations of HMLN-1 and *L. reuteri* M4-100 strains as those used in the adhesion and invasion assays were inoculated into each insert. Inhibition of translocation was measured by adding both M4-100 and HMLN-1 simultaneously (co-inoculation assay) before incubation at 37 °C under 5% CO_2_ for 60 min. The pre-inoculation assay was conducted by incubating the lactobacillus strain for 60 min prior to HMLN-1 inoculation and then further incubated at 37 °C under 5% CO_2_ for 60 min. After incubation, 100 µL of EMEM was taken from the outer well, serially diluted, and inoculated onto MacConkey agar plates. Plates were incubated overnight and counted for CFU, and the results are expressed as mean ± SEM. All translocation experiments were performed in triplicate. The *E. coli* strain JM109 was used as a negative control.

### 2.6. Scanning Electron Microscopy (SEM)

Interactions of HMLN-1 with the co-culture cells were recorded using a scanning electron microscope. The cells of co-culture were grown in Millicell inserts (Millipore, Australia) in a 24-well tissue culture plate as described above. Inserts were seeded with 400 µL of growth media containing (4 × 10^5^ CFU/mL) of Caco2:HT29-MTX and incubated at 37 °C in 5% CO_2_ until confluent. Cells were then inoculated with a suspension of HMLN-1 (~10^7^ CFU/mL^−1^). Following a 60- and 120-min incubation period, the monolayer within the inserts was washed with PBS and fixed in situ with 2.5% glutaraldehyde in PBS overnight at 4 °C, followed by heavy metal staining with 1% osmium tetraoxide in 0.1 M sodium cacodylate buffer to ensure the staining of structural cellular components. Dehydration was completed using progressively increasing concentrations of ethanol at 10%, 20%, 30%, 40%,50%, 60%, 70%, 80%, 90%, and 100% (twice). Samples were sputter-coated using a platinum coater for a thickness of 4 nm (Leica systems). Imaging was performed with an acceleration voltage of 5 kV and beam intensity of 8 using a Tescan MIRA 3 Scanning Electron Microscope (Brno, Czeck Republic)

### 2.7. Transmission Electron Microscopy (TEM)

The sample protocol as described above for SEM was also employed for TEM. Dehydration was carried out through a graded ethanol series at 10%, 20%, 30%, 40%, 50%, 60%, 70%, 80%, 90%, and 100%, followed by embedding into EPON812-substitute resin through a graded ethanol-to-resin series of 25%, 50%, 75%, and 100%, followed by curing at 60 °C. The processed samples were ultra-thin-sectioned using a Leica UC7 Ultramicrotome (Leica, Vienna, Austria) to 70 nm, and post-stained with 2% uranyl acetate and lead citrate. TEM imaging was conducted on a JEOL 1400 TEM (JEOL, Tokyo, Japan), operated at 120 kV, mounted with a TVIPS F216 2K CCD camera (Puchheim, Germany).

### 2.8. RNA Extraction

To identify differential gene expression in Caco2:HT29-MTX cells upon the interaction with the HMLN-1 and *L. reuteri* M4-100, the co-culture was grown in 12-well microplates as described above until reaching complete confluency. RNA was extracted from the following experimental groups: Group 1 (Caco2:HT29-MTX cells alone), Group 2 (HMLN-1 plus Caco2:HT29-MTX cells), and Group 3 (*L. reuteri* M4-100 plus Caco2:HT29-MTX cells). Each experiment was completed in triplicate. After exposure of the co-culture to the respective treatments, the cells were incubated for 60 min, after which the culture medium was removed, and the cells were washed with PBS. Cells were then trypsinized with a 0.25% trypsin–EDTA (Thermo Fisher Scientific, Banyo, Qld, Australia) solution to detach them from the 12-well plates. The reaction was then stopped by adding culture medium containing serum, and the cells were collected by centrifugation at 500× *g* for 5 min at 4 °C. The supernatant was discarded, and the cell pellets were resuspended in RNA lysis buffer (RLT buffer with 1% β-mercaptoethanol) to ensure RNA preservation. The lysates were homogenized using a mechanical homogenizer for effective disruption. RNA was extracted from the cell pellets using a commercially available RNA extraction kit following the manufacturer’s protocol (Qiagen RNeasy Mini Kit, Clayton, Victoria, Australia). The quantity and purity of RNA were checked using spectrophotometry (NanoDrop, ThermoFisher) and gel electrophoresis, which was performed using 1.5% agarose gel containing formaldehyde under denaturing conditions at 100 V for 45 min, and RNA was visualized under UV illumination. The samples were then sent to a commercial service, the Australian Genome Research Facility (AGRF), for RNA sequencing. Library preparation, cDNA synthesis, and sequencing were performed on an Illumina platform as per the AGRF service guidelines. Image analysis was performed in real time by NovaSeq Control Software (NCS) v1.2.2.48004 and Real Time Analysis (RTA) v4.6.7, running on the instrument computer. The Illumina DRAGEN BCL Convert 07.031.732.4.3.6 pipeline was used to generate the sequence data. Paired-end reads of 150 bp were used for high-throughput sequencing to ensure the accurate representation of the transcriptome. The data yield for each sample was in the range of 10–15 gigabases.

### 2.9. RNA-Seq Data Analysis

Gene expression analysis was conducted using CLC Genomics Workbench version 21.0.3 (QIAGEN Bioinformatics, Aarhus, Denmark). Raw sequencing reads were quality-filtered using the Trim Sequences tool in the CLC Genomics Workbench. This included automatic removal of adapter sequences, trimming of low-quality bases with a quality score limit of 0.05, exclusion of reads containing more than two ambiguous nucleotides, and removal of sequences shorter than 15 nucleotides after trimming. These steps ensured the generation of high-quality data for downstream analysis. The trimmed reads were then mapped to the reference genome *Homo sapiens* GRCh38.p14 using the RNA-Seq Analysis tool and CDS databases for annotation. Gene expression levels were quantified by calculating the number of reads mapped to each gene and normalized using the RPKM (Reads Per Kilobase of transcript per Million mapped reads) method. Differential expression analysis was performed using the Differential Expression for RNA-Seq tool, comparing treatment groups under default statistical settings (including TMM normalization and exact test for two-group comparisons). Significantly differentially expressed genes (DEGs) were identified based on adjusted *p*-values (FDR < 0.05). The normalized gene expression data were exported and used to generate heatmaps using SRplot (https://www.bioinformatics.com.cn/en, accessed on 12 June 2025), which visualized expression profiles across all experimental conditions. Gene clusters from the heatmap were further analyzed for functional enrichment and biological pathways.

## 3. Statistical Analysis

Statistical analysis was conducted using GraphPad Prism (Version 8.0.0 GraphPad Software, San Diego, CA, USA). Differences in mean levels of adhesion, invasion, and translocation among strains across all test groups were assessed using two-way ANOVA followed by Tukey’s multiple comparisons test. Differences were considered statistically significant if *p* < 0.05.

## 4. Results

### 4.1. Scanning Electron Microscopy (SEM)

The HMLN-1 strain showed a high ability to colonize the co-culture cells in both co-inoculation and in pre-inoculation experiments. Scanning electron microscopy also showed that the strain closely adhered to the microvilli of the co-culture cells (Figure 1).

### 4.2. Adhesion Assay

More than 60% of the cells in the co-culture were colonized with HMLN-1. However, these values were reduced in both the co-inoculation and pre-inoculation experiments by 15% and 68%, respectively, when *L. reuteri* M4-100 was present (Figure 2). The mean number of HMLN-1 adhering per cell alone (6.8 ± 0.7 CFU/cell) in the co-inoculation (4.3 ± 0.8) and pre-inoculation (1.4 ± 0.2) experiments with *L. reuteri* M4-100 was significantly (*p* < 0.0001) reduced. A highly significant reduction (*p* < 0.0001) was also found in the invasion of the co-culture by the HMLN-1 strain when it was co-inoculated (65.6%) and pre-inoculated (97%) with *L. reuteri* M4-100 (Figure 3). In general, the pre-inoculation of *L. reuteri* M4-100 was always associated with a greater reduction than co-inoculation.

### 4.3. Invasion Assay

A highly significant reduction (*p* < 0.0001) was also found in the invasion of the co-culture by the HMLN-1 strain when it was co-inoculated (65.6%) and pre-inoculated (97%) with *L. reuteri* M4-100 (Figure 3). In general, the pre-inoculation of *L. reuteri* M4-100 was always associated with a greater reduction than co-inoculation.

### 4.4. Translocation Assay

Both HMLN-1 and the positive control (F44A-1 strain) translocated efficiently in the co-culture cells at concentrations of 1 × 10^6^ and 4.5 × 10^6^ CFU/well, respectively; however, these values were significantly (*p* < 0.0001) reduced in the presence of *L. reuteri* M4-100. Pre-inoculation of the co-culture showed an even further reduction of translocation (*p* < 0.0001) than the co-inoculation (Figure 4).

### 4.5. Transmission Electron Microscopy (TEM)

Transmission electron microscopy of the co-culture cells after 120 min incubation showed an initial adhesion of the HMLN-1 to microvilli, followed by the internalization of the cell by a membrane-bound vesicle and passage through the cell (Figure 5).

### 4.6. Differential Gene Expression Results

The differential gene expression analysis of Caco2:HT29-MTX cells following sixty minutes of direct contact with HMLN-1 and *L. reuteri* M4-100 revealed distinct immune and epithelial barrier responses compared to the control group in three major functional categories: pro-inflammatory response, epithelial barrier integrity, and anti-inflammatory regulation. Differential expression of genes within these categories was selected based on their statistically significant values determined by FDR (False Discovery Rate) values (Table 1). HMLN-1 exposure triggered a strong pro-inflammatory response, with a significant upregulation of NF-kappa-B inhibitor zeta (NFKBIZ), tumor necrosis factor (TNF), and toll-like receptor 7 (TLR7). Additionally, a mucin gene, MUC1, was also upregulated upon exposure to HMLN-1, as were several genes involved in the anti-inflammatory response, such as IL2RA, NFKBIA, NFKB2, and TNFRSF19. In contrast, exposure to *L. reuteri* M4-100 resulted in the downregulation of pro-inflammatory gene IL1B and TGFB1 and the anti-inflammatory response gene (Table 1).

### 4.7. Gene Expression Clustering Analysis Results

A hierarchical clustering heatmap was generated based on the differentially expressed genes following exposure to the translocating *E. coli* strain HMLN-1, probiotic *L. reuteri* M4-100, and the untreated control group (i.e., human epithelial cell line alone) (Figure 6). Gene expression values were normalized and color-coded, with red indicating upregulation and blue indicating downregulation relative to the mean expression level. The results reveal three distinct clusters corresponding to each treatment group, indicating specific gene expression signatures induced by each treatment.

Figure 6 shows the independent clustering of treatments, displaying minimal transcriptional variation. The control group (green) reflects the baseline gene expression profile of the human intestinal epithelial cells without any bacterial exposure, whereas the HMLN-1 group (pink) represents cells exposed to the translocating *E. coli* strain, and the M4-100 group (blue) displays cells that have been treated with the probiotic strain.

Exposing the human intestinal epithelial cells to bacteria (whether pathogenic or probiotic) caused a dramatic shift in gene expression in gene categories associated with inflammatory response and barrier integrity. Differences were also observed between the pathogenic HMLN-1 strain and the probiotic M4-100 strain. The HMLN-1 cluster showed a prominent upregulation of pro-inflammatory genes, such as TNF, CXCL3, and NFKBIE. Conversely, anti-inflammatory and regulatory genes, like TGFB1 and IL1B, were downregulated. The M4-100 group exhibited a contrasting gene expression profile characterized by an upregulation of RAB27B involved in exosome secretion, and upregulation in genes, such as MUC13, TNFSF15, and TACC1. Furthermore, MUC5B was upregulated upon inoculation with the pathogenic HMLN-1 strain, while probiotic M4-100 exposure resulted in its downregulation (Figure 6).

To explore the host’s transcriptional response to *Escherichia coli* HMLN-1 and the regulatory impact of *L. reuteri* M4-100, RNA sequencing data were analyzed using the CLC Genomics Workbench. Genes showing a differential expression were identified using the criteria of an FDR-adjusted *p*-value below 0.05 and an absolute log_2_ fold change (log_2_ FC) of at least 2. Volcano plots (Figure 7) illustrate the patterns of differentially expressed genes (DEGs) under each condition. In Figure 7A, it can be seen that exposure to HMLN-1 caused extensive changes in gene expression in epithelial cells, with many genes significantly upregulated (blue) or downregulated (red), reflecting a pronounced proinflammatory or pathogenic response. Conversely, Figure 7B displays the gene expression profile following treatment with the probiotic M4-100 strain.

Genes represented by gray dots in both plots had either low fold changes (log_2_ FC < 2) or were not statistically significant (FDR > 0.05).

## 5. Discussion

The ability of probiotics to compete with pathogens for binding sites has led to increased research into their potential to prevent disease from pathogenic bacteria [17]. Multiple studies have found that probiotics can effectively reduce the number of pathogenic bacteria adhering to enterocytes; additionally, more recent research has determined that probiotics can also reduce the invasion and translocation of pathogens across intestinal epithelium, [16,17,20]. However, it must be noted that many studies have been performed on single-cell line monolayers, such as Caco-2, which may not fully represent the components of the epithelial lining of the gut. Despite their likeness to enterocytes, Caco-2 cells lack metabolizing enzymes and specific transporter expression, although they form an effective mechanical barrier by producing tight junctions (TJs) [27,29,30].

One of the important limitations of Caco-2 cells as a model of the gut epithelium is their inability to produce mucin. HT29-MTX cells on the other hand have the same morphology as HT-29 cells, but over time, they can secrete mucin, specifically MUC2, on their apical axis, which simulates a mucosal membrane layer [26]. By combining these cells in a co-culture, as used in this study, it is possible to mitigate some of these limitations and create an effective model of the human intestinal epithelial layer [12,15]. The efficacy of combining Caco-2 and HT29-MTX cell lines has been widely documented and is currently the most common method for replicating the GI epithelium [12,25,31]. In this improved model of the intestinal epithelium, the HMLN-1 strain showed a high ability to adhere, invade, and translocate in this co-culture of cells. Scanning electron microscopy showed the adhesion of HMLN-1 to the apical axis of the microvilli of the cells. Translocation was also showed to be an intracellular process, where the adhesion of the HMLN-1 strain was followed by its internalization into a membrane-bound vesicle to be translocated through the epithelial cells, rather than via the paracellular pathway. This result was further supported by steady TEER readings throughout the assay. A similar process has also been reported for other pathogenic bacteria [22,31]. For instance, *Salmonella enterica* serovar Typhimurium invades intestinal epithelial cells by inducing its own uptake into membrane-bound vacuoles, allowing translocation through the cells [32]. Similarly, *Listeria monocytogenes* enters epithelial cells and utilizes an intracellular route to cross the intestinal barrier [33]. Additionally, *Shigella flexneri* invades colonic epithelial cells and translocates intracellularly, avoiding the disruption of tight junctions [34]. These pathogens, like HMLN-1, exploit the host cell machinery to facilitate internalization and intracellular trafficking, supporting the concept that bacterial translocation can occur via intracellular mechanisms without compromising epithelial barrier integrity, as indicated by stable TEER measurements.

As in the above pathogens, HMLN-1 also carries T6SS genes; but, in this study, we not only provided evidence of the interaction of HMLN-1 with the intestinal epithelium, including its route of translocation, we also used an improved model of the intestinal epithelium, where the cell lines were covered with mucin, a substance that acts as the gatekeeper of the mucosal barrier and controls or deters invading pathogens of the epithelial cells [35]. This was different to previous studies in which one cell type was used [20,36]. Here, we also showed a high level of expression of genes involved in inflammatory response and tight junction integrity after the interaction with HMLN-1 in our model of the intestinal epithelium.

Probiotics have been shown to have several beneficial effects, including improvement of intestinal health, as well as their use in the treatment of diarrheal diseases, enhancement of the immune response, and cancer prevention [18,19]. These properties, however, are strain-specific and are impacted by various mechanisms, such as the production of bacteriocin and short-chain fatty acids, competition for nutrients, and stimulation of mucosal barrier function and immunomodulation [37,38]. In our study, *L. reuteri* M4-100 exhibited an ability to colonize the co-culture of cells comparable to the HMLN-1 strain, and reduced the adherence of this strain both in the co-inoculation and pre-inoculation experiments. However, we did not find any correlation between the extent of *L. reuteri* M4-100’s colonization, its adhering number per cell, and its ability to reduce the adhesion of the HMLN-1 strain. This suggests that the competitiveness of probiotic strains against the pathogens might not be solely due to the their direct competition for binding sites, and other factors, such as the secretion of antimicrobial substances, could be involved in their competitive efficacy, as postulated by others [36,39]. It has also been shown that *Lactobacillus* strains over a 24 h period could not inhibit the adhesion of adherent–invasive *E. coli* (AIEC) to mucus, but it did impact the growth and survival of the pathogen [22]. This indicates that the observed reduction in adhesion in this study and in other studies occurs at the epithelial layer rather than in the mucus compartment [40,41].

Despite the highly invasive nature of the HMLN-1 strain, the *L. reuteri* M4-100 strain significantly reduced its invasion, which corresponded well with the findings from the adhesion assays, where the *L. reuteri* M4-100 strain was demonstrated to be a potent inhibitor of HMLN-1 adhesion. Similar observations have been reported for other probiotic bacteria, where their antagonistic effects against pathogens extend beyond simple competition for adhesion sites. For example, certain *Lactobacillus* and *Bifidobacterium* strains have been shown to inhibit pathogens like *E. coli*, *Salmonella*, and *Clostridium difficile* through the secretion of antimicrobial peptides, production of organic acids, and modulation of host immune responses rather than directly blocking adhesion [42,43]. Moreover, *Lactobacillus rhamnosus* GG was found to reduce the adhesion of enteropathogenic *E. coli* to epithelial cells by enhancing epithelial barrier function and inducing the expression of mucins and defensins, rather than solely occupying adhesion sites [44]. These findings support the concept that probiotic efficacy involves multifactorial mechanisms, including antimicrobial activity and host modulation, consistent with the observations from our study and others.

The transcriptomic analysis of the intestinal epithelial cells in this study (Caco2:HT29-MTX co-culture) revealed distinct gene expression patterns in response to pathogenic *E. coli* HMLN-1 and the effect of *L. reuteri* M4-100. In human epithelial cells alone, various genes are naturally up- or downregulated, representing baseline immune homeostasis. However, following exposure to either HMLN-1 or *L. reuteri* M4-100, the expression of key inflammatory, immune modulatory, and epithelial barrier-related genes was significantly altered, highlighting the dynamic interaction between host cells and microbes. The heatmap visualization of the RNA sequencing data (Figure 6) highlights the differential gene expression patterns of intestinal epithelial cells exposed to HMLN-1 and the probiotic *L. reuteri* M4-100 strains as opposed to untreated control cells. Exposure to *E. coli* HMLN-1 resulted in a strong upregulation of several proinflammatory and immune-related genes, including TNF, TLR7, and NFKBIZ, as well as genes involved in the anti-inflammatory response, such as NFKBIA, NFKB2, and TNFRSF19. Similar gene expression profiles have been observed in host responses to other pathogenic or translocating bacteria, such as *S. enterica, Campylobacter jejuni*, and adherent–invasive *E. coli* [32,45,46]. For example, TLR7, NFKBIA, NFKBIZ, NFKB2, and TNFRSF19 have been shown to be upregulated in epithelial cells and macrophages during infection with AIEC strains isolated from Crohn’s disease patients [47]. These genes are central regulators of the NF-κB and TNF signaling pathways, which control the balance between proinflammatory activation (TNF, NFKBIZ) and negative feedback or immune resolution (NFKBIA, NFKB2, TNFRSF19).

The simultaneous induction of both pro- and anti-inflammatory genes suggests a dysregulated or unresolved inflammatory response, a hallmark of chronic inflammation. Such expression patterns are commonly associated with early stages of diseases, like inflammatory bowel disease (IBD), including Crohn’s disease and ulcerative colitis, and may also contribute to epithelial barrier dysfunction, bacterial translocation, and even tumorigenesis in colorectal cancer [48,49].

Furthermore, differential expression patterns observed between the control group and exposure to either bacterial strain in the heatmap indicate that mucin-related genes, such as MUC1, MUC13, and MUC5B, become upregulated, potentially indicating an epithelial defense mechanism aimed at reinforcing the mucosal barrier. In contrast, bacterial exposure led to the downregulation of genes involved in immune regulation, such as TGFB1 [50], and interleukin 1 beta (IL-1β), a potent pro-inflammatory cytokine [51] potentially affecting the activation of neutrophils and macrophages. A noteworthy difference between exposure to *E. coli* strain HMLN-1 and treatment with the probiotic strain *L. reuteri* M4-100 was observed in the opposing gene expression of MUC5B (highly upregulated with the pathogenic strain but downregulated with the probiotic strain) and RAB27B (downregulated with the HMLN-1 but upregulated with the probiotic strain). MUC5B was found to be upregulated in patients with chronic rhinosinusitis and associated with a higher prevalence of bacterial biofilms [52] indicating a possible mechanism for bacterial pathogenesis. RAB27B was also found to be involved in expelling intracellular, uropathogenic *E. coli* (UPEC) from its intracellular niche [53]. This function may present an important mechanism of action for probiotics in the host response.

In conclusion, this study found that the HMLN-1 strain adhered to microvilli and internalized into Caco-2:HT29-MTX cells via membrane-bound vesicles. The probiotic strain *L. reuteri* M4-100 effectively reduced the interaction of HMLN-1 with our intestinal epithelial cell model, with the most significant effect observed when *L. reuteri* M4-100 was pre-inoculated 90 min before HMLN-1 exposure, suggesting its potential as a prophylactic measure. Additionally, HMLN-1 induced a strong inflammatory response and compromised intestinal epithelial integrity, while *L. reuteri* M4-100 counteracted these effects by enhancing barrier function and regulating immune responses. Heatmap analysis highlighted contrasting gene expression patterns between human cells with and without bacterial exposure, and between pathogenic and probiotic treatments. This underscores the potential of *L. reuteri* M4-100 as a probiotic intervention to restore immune balance and protect gut epithelial integrity from the invasion and translocation of pathogenic bacteria.

## Figures and Tables

**Figure 1 microorganisms-13-01428-f001:**
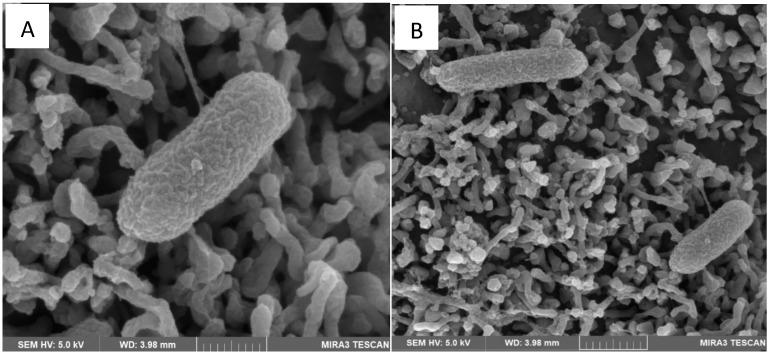
Scanning electron micrograph showing adhesion of the HMLN-1 strain to the microvilli of the co-culture of Caco-2:HT29-MTX cells after 120 min incubation. Scale bar: 500 nm (**A**) and 1 µm (**B**).

**Figure 2 microorganisms-13-01428-f002:**
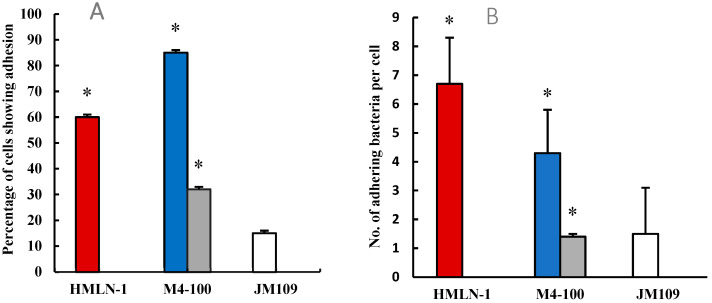
(**A**) The percentage colonization of the co-culture of Caco-2:HT29-MTX cells by the HMLN-1 alone 
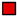
 and in the presence of the *Limosilactobacillus* strain when co-inoculated 
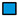
 and pre-inoculated 
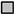
. *E. coli* strain JM109 used as the negative control 
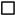
. (**B**) The number (mean ± SEM) of adhering HMLN-1 
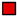
 alone and in the presence of the M4-100 when co-inoculated 
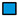
 and pre-inoculated 
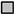
. *E. coli* strain JM-109 
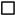
 was used as a negative control. * *p* < 0.0001 co-inoculation versus pre-inoculation in both (**A**,**B**) graphs.

**Figure 3 microorganisms-13-01428-f003:**
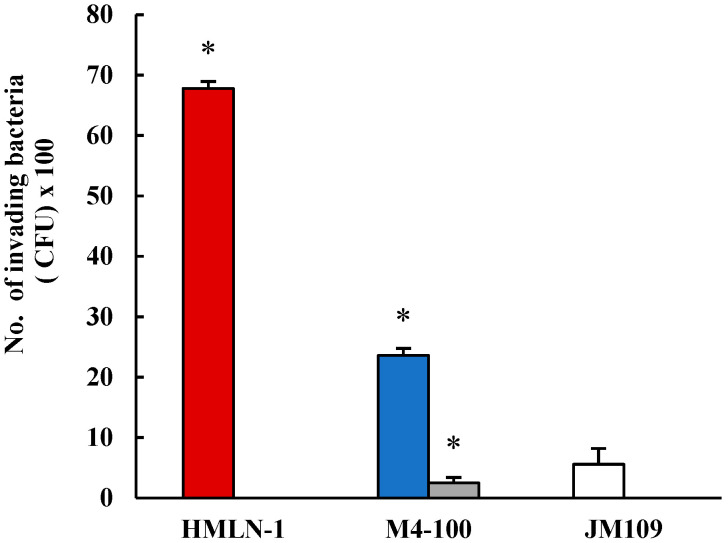
The number (mean ± SEM) of invading HMLN-1 (CFU) alone 
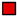
, in the presence of M4-100 when co-inoculated 
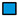
 and pre-inoculated. 
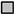

*E. coli* strain JM-109 used as the negative control 
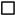
. * *p* < 0.0001 co-inoculation versus pre-inoculation.

**Figure 4 microorganisms-13-01428-f004:**
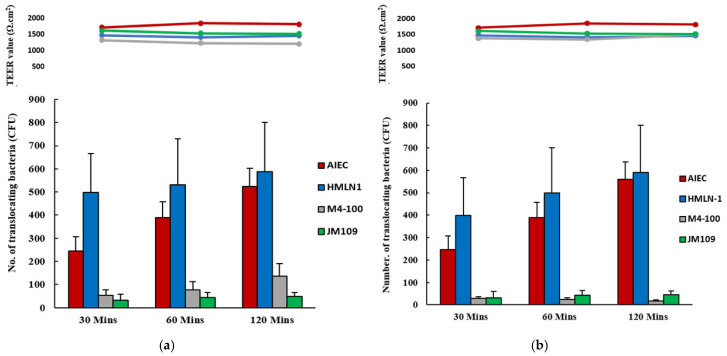
The number of translocating HMLN-1 strains alone 
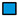
, and in the presence 
of *L. reuteri* M4-100 
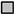
 in co-inoculation (**a**) and in pre-inoculation experiments (**b**). *E. coli* strain F44A-1, an AIEC strain, was used as a positive control 
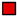
 and *E. coli* strain JM109 used as a negative control 
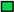
. Translocation was measured after 30, 60, and 120 min. Results are expressed as mean ± SEM.

**Figure 5 microorganisms-13-01428-f005:**
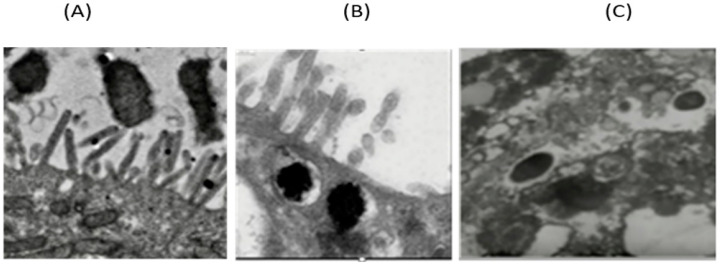
Transmission electron microscope showing (**A**) the adhesion of HMLN-1 to microvilli, (**B**) internalization by a membrane-bound vesicle, and (**C**) passage through the cell after 120 min incubation.

**Figure 6 microorganisms-13-01428-f006:**
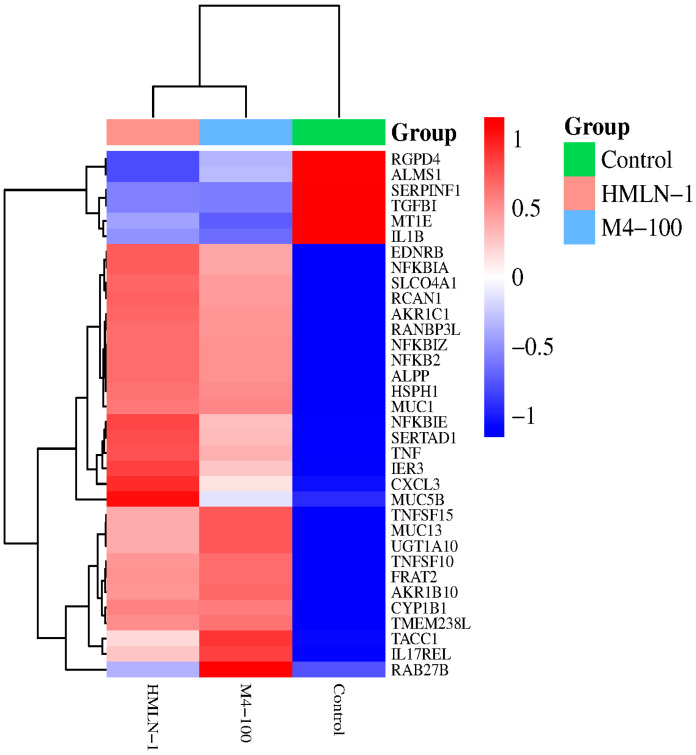
Heatmap showing all significantly differentially expressed genes across conditions (red indicates gene upregulation while blue shows downregulation).

**Figure 7 microorganisms-13-01428-f007:**
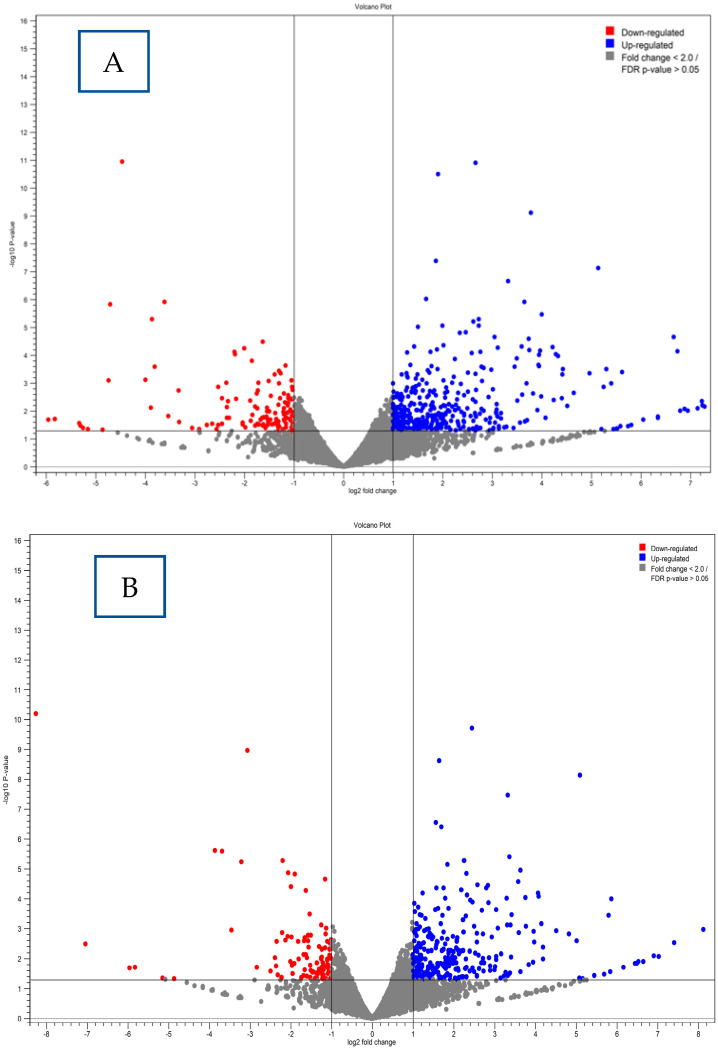
Volcano plots of differential gene expression in epithelial cells. Panel (**A**) reflects Caco2:HT29-MTX exposure to HMLN-1, while panel (**B**) shows the effect of treatment with the probiotic M4-100 strain.

**Table 1 microorganisms-13-01428-t001:** Differentially expressed genes involved in inflammatory responses and tight junction pathways with their corresponding FDR values. Upregulation ↑; downregulation ↓.

Gene Category	Gene Name	FDR Value	
		HMLN-1 vs. Control	*L. reuteri* M4-100 vs. Control
Genes involved in pro-inflammatory response	IL1B		↓ 0.001895
	IL17REL		↑ 0.004429
	NFKBIZ	↑ 4.25E-08	↑ 3.9E-07
	IL6ST		
	TNF	↑ 0.000455	↑ 0.001208
	TLR7	↑ 0.00033	↑ 0.000236
Genes involved in tight junction and adherence	L1CAM		
	MUC13	↑ 0.00313096	↑ 0.002085
	MUC6		
	MUC5B		
	MUC1	↑ 0.00313084	↑ 0.004085
	MAPK4		
Genes involved in anti-inflammatory response	TGFB1		↓ 0.00546
	CXCL3	↑ 0.003130	↑ 0.004085
	IL2RA	↑ 0.009759	
	NFKBIA	↑ 3.2E-11	↑ 2.37E-09
	NFKB2	↑ 0.009517	
	TNFRSF19	↑ 7.59E-05	↓ 0.000132
	CXCL5		↑ 0.009013

## Data Availability

The original contributions presented in this study are included in the article. Further inquiries can be directed to the corresponding author.

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
