# Peer review of "Limosilactobacillus reuteri M4-100 Mitigates the Pathogenicity of Escherichia coli Strain HMLN-1 in an Intestinal Epithelial Model and Modulates Host Cell Gene Expression"

_microorganisms, 2025, doi:10.3390/microorganisms13061428_

Round 1

Reviewer 1 Report

Comments and Suggestions for Authors

The manuscript addresses an important and timely topic—the role of probiotics, specifically Limosilactobacillus reuteri M4-100, in mitigating pathogenic E. coli HMLN-1 interactions with intestinal epithelial cells. The integration of co-culture epithelial models, functional assays (adhesion, invasion, translocation), microscopy, and transcriptomic profiling represents a thorough and multidimensional experimental design. However, while the experimental setup is largely solid and the writing is clear, there are several concerns regarding methodological clarity, depth of interpretation, and novelty that must be addressed prior to publication.

Major Comments

  1. Novelty and Impact
    While the findings reinforce the probiotic potential of L. reuteri M4-100, the manuscript lacks a strong emphasis on what is new compared to previous studies, including the authors’ own prior work (Refs 20, 21). The manuscript should better articulate:

    • What distinguishes this study from previous probiotic-E. coli interaction studies?

    • Is HMLN-1 mechanistically different from other AIEC strains previously studied?

  2. Lack of Mechanistic Insights
    The study shows reduction in adhesion/invasion/translocation and altered gene expression, but does not clarify the mechanism of probiotic action (e.g., competition for receptors, secretion of antimicrobial compounds, modulation of signaling). This significantly limits the biological interpretation of the findings.

  3. RNA-Seq Interpretation Needs Improvement

    • The RNA-seq section is technically sound, but the discussion of gene function is superficial. For example, the mention of IL1B and TGFB1 regulation lacks insight into how this reflects the probiotic's mechanism.

    • Key pathways (e.g., NF-κB, mucin biosynthesis, TLRs) should be mapped and discussed with proper references or pathway diagrams.

  4. Figures and Data Presentation

    • Figures (particularly heatmaps and microscopy) are described in the text but are not shown in the version provided. The quality and clarity of these figures are essential for evaluating the claims.

    • It is unclear how gene expression changes correlate with functional outcomes (e.g., decreased translocation). Were any genes involved in tight junction formation or vesicular trafficking highlighted?

Minor Comments

  1. Terminology Consistency

    • The authors refer to Caco-2 and HT29-MTX co-cultures, sometimes as “gut epithelium”, “intestinal model”, or “epithelial barrier”. It would help to define the model clearly at first mention and use consistent terminology.

  2. Statistical Clarifications

    • The number of biological replicates (not just technical replicates) for adhesion, invasion, and RNA-seq should be explicitly stated.

    • All p-values are marked <0.0001; were adjusted p-values (FDR) applied in microbiological assays, or only in transcriptomics?

  3. Experimental Controls

    • It is commendable that both positive (AIEC strain F44A-1) and negative (JM109) controls were used. However, inclusion of a non-probiotic Lactobacillus strain could help clarify if the observed effects are strain-specific.

  4. Materials and Methods

    • Several procedures (e.g., RNA quality assessment, sequencing depth, gene filtering criteria) are described in broad terms. More detail is needed, particularly in sections 2.8 and 2.9.

  5. Discussion Needs Structuring

    • The discussion often repeats the results and jumps between topics. It would benefit from restructuring around key themes: (i) HMLN-1 pathogenicity, (ii) probiotic mitigation effects, (iii) gene expression modulation, (iv) implications for probiotic therapy.

Suggestions for Improvement

  • Clarify novelty over previous studies and reinforce contributions to the field.

  • Expand on mechanistic hypotheses and, if possible, test whether M4-100 secretes antagonistic metabolites.

  • Improve integration of transcriptomic data with functional results.

  • Include pathway analyses and biological interpretation of key regulated genes.

  • Revise and reorganize the discussion section for clarity and coherence.

Recommendation: Major Revision

The study is scientifically sound and well-conducted, but requires revision to clearly present the novelty, improve mechanistic interpretations, and more thoroughly discuss transcriptomic results. Upon addressing these concerns, the manuscript may be suitable for publication.

Reviewer 2 Report

Comments and Suggestions for Authors
  • It appears that the reduction in E. coli adhesion by L. reuteri may be achieved through competition for binding sites on the epithelial cells. To further clarify this mechanism, the authors are encouraged to include experiments using heat-killed L. reuteri to determine whether the observed effects are due to live bacterial metabolism or merely physical occupancy. Such experiments would help assess whether secreted metabolites or surface structures play a regulatory role in E. coli adhesion.

  • In Figure 6, the authors present a heatmap as the sole output of transcriptomic analysis, which provides limited insight without supporting statistical and dimensionality reduction analyses. It is unclear whether the data represent biological replicates or a single experiment. The authors should strengthen this section by including a volcano plot, PCA (Principal Component Analysis), and Venn diagrams to illustrate differential gene expression more comprehensively. Moreover, in-depth enrichment analyses such as KEGG and GO would be valuable in elucidating the pathways involved in E. coli adhesion and host cell interaction.

Reviewer 3 Report

Comments and Suggestions for Authors

The authors investigated the ability of L. reuteri M4-100 to competitively inhibit invasion and translocation of the TEC strain (HMLN-1) and the mechanisms by which the HMLN-1 strain interacted with the intestinal epithelium and its translocation pathway. Gene expression studies were carried out, demonstrating key gene categories associated with the host response to infection with this strain and the molecular effects of exposure to the probiotic L. reuteri M4-100 in an intestinal epithelial cell culture model.

As intestinal opportunistic pathogens E. coli  could translocate from the intestinal tract to mesenteric lymph nodes and portal system, and then to other external tissues or organs in specific ways, which could cause a range of serious diseases, such as sepsis, meningitis, bacteremia, necrotizing small bowel colitis, mastitis, etc. Currently, antibiotics are a primary approach against pathogens, but their efficacy has decreased due to the emergence of resistant pathogens. To date, numerous evidence has suggested that probiotics could prevent bacterial translocation by maintaining intestinal homeostasis and enhancing intestinal defense, indicating the potential of probiotics to replace antibiotics in pathogenic infection treatment. I therefore consider the research carried out to be very important and necessary.

There are too many self-citations of the authors' own research in this manuscript, as many as 10. Especially the large descriptions of the authors' own research in the introduction. I would suggest citing other authors' studies as well, this would greatly enrich the manuscript.

The authors should also clarify the aim of the study - as it stands, they have only given the scope of the study.

It is a pity that the tolerance of the tested bacteria to simulated digestion conditions (in gastrointestinal fluid) was not investigated. An interesting study would have been to determine antibiotic sensitivity and haemolytic activity.

In these studies, the expression of key inflammatory, immunomodulatory and epithelial barrier-related genes was significantly altered, highlighting the dynamic interaction between host cells and microorganisms. Exposure to E. coli HMLN-1 caused, strong up-regulation of several pro-inflammatory and immune-related genes: TNF, TLR7, L17REL AND NFKBIZ for M4-100. Such strong activation of the pro-inflammatory pathway, is not a good prognosis for intestinal epithelial health. This problem should be described more extensively in the discussion and the findings of other authors should be referred to in more depth.

Round 2

Reviewer 2 Report

Comments and Suggestions for Authors

I agree that this article could be accepted

Reviewer 3 Report

Comments and Suggestions for Authors

The authors have revised the manuscript. I have no further comments.